# Elastin-like Polypeptide Hydrogels for Tunable, Sustained Local Chemotherapy in Malignant Glioma

**DOI:** 10.3390/pharmaceutics14102072

**Published:** 2022-09-28

**Authors:** Sonja Dragojevic, Lindsay Turner, Pallabi Pal, Amol V. Janorkar, Drazen Raucher

**Affiliations:** 1Division of Radiation Oncology, Mayo Clinic and Foundation, 200 First Street, Rochester, MN 55905, USA; 2Department of Cell and Molecular Biology, University of Mississippi Medical Center, 2500 North State Street, Jackson, MS 39216, USA; 3Department of Biomedical Materials Science, School of Dentistry, University of Mississippi Medical Center, 2500 North State Street, Jackson, MS 39216, USA

**Keywords:** sustained drug delivery, doxorubicin, elastin-like polypeptide, glioblastoma, hydrogels

## Abstract

Glioblastoma (GBM) is a primary brain tumor that carries a dismal prognosis, which is primarily attributed to tumor recurrence after surgery and resistance to chemotherapy. Since the tumor recurrence appears near the site of surgical resection, a concept of immediate and local application of chemotherapeutic after initial tumor removal could lead to improved treatment outcome. With the ultimate goal of developing a locally-applied, injectable drug delivery vehicle for GBM treatment, we created elastin-like polypeptide (ELP) hydrogels. The ELP hydrogels can be engineered to release anti-cancer drugs over an extended period. The purpose of this study was to evaluate the biomechanical properties of ELP hydrogels, to characterize their ability to release doxorubicin over time, and to investigate, in vitro, the anti-proliferative effect of Dox-laden ELP hydrogels on GBM. Here, we present microstructural differences, swelling ratio measurements, drug release characteristics, and in vitro effects of different ELP hydrogel compositions. We found that manipulation of the ELP–collagen ratio allows for tunable drug release, that the released drug is taken up by cells, and that incubation with a small volume of ELP-Dox hydrogel drastically reduced survival and proliferation of GBM cells in vitro. These results underscore the potential of ELP hydrogels as a local delivery strategy to improve prognosis for GBM patients after tumor resection.

## 1. Introduction

Glioblastoma (GBM) is the most common and aggressive primary malignant brain tumor in the U.S., with average annual age-adjusted incidence rate of 59,000 cases in the time period between 2012 and 2016 [1] carrying a bleak prognosis. Following the initial diagnosis of all malignant CNS tumors, a five-year survival rate for GBM patients accounts for only 6.8% of patients [1]. Upon diagnosis of GBM, a patient would undergo maximal surgical resection followed by radiotherapy and concomitant chemotherapy with temozolomide [2]. The implementation of gold standard therapy in 2005 resulted in extended median patient survival of three months on average. The modest improvement in overall survival still represents a 25% increase in survival time [3]. Surgical reduction of tumor mass generally does not result in complete tumor resection in GBM due to poorly defined tumor borders. Residual tumor cells proliferate rapidly, and tumor recurrence at or adjacent to the margins of the resection area is common [4]. Post-surgical treatment with oral temozolomide and radiation has shown to improve overall survival [5]; however, anticancer drugs administered systemically have limited effectiveness in tumor regions with an intact blood–brain barrier (BBB), which prevents drug delivery to the cancer cells. One approach to improve low BBB penetration is through a delivery system based on receptor-mediated transcytosis using monoclonal antibody conjugates [6]. However, this approach is still in development and new approaches are urgently needed. In addition to low BBB penetration of the chemotherapeutics, patients with glioblastoma are often negatively impacted by side effects of treatment. With very limited clinically approved treatment regimens and numerous unwanted side effects, there is an unmet need for additional treatment options as well as better balance of tumor–healthy tissues exposure to achieve efficacious concentrations within the tumor while limiting the healthy tissue exposure to cytotoxic compounds, that put additional limits on systemically administered chemotherapy. Furthermore, since radiation and chemotherapy retard the wound healing process after resection, conventional treatment for GBM patients is delayed for several weeks after surgery. During this delay between surgery and the start of chemo and radiotherapy, residual cancer cells have time to proliferate and cause recurrence. As evidenced by the grim survival statistics for GBM, new treatment options, including a smart and adaptable drug delivery systems, are urgently needed to improve patient outcomes.

The BBB is important for maintaining CNS equilibrium, while it also presents a major obstacle in efficient drug delivery to the brain. The impenetrable wall of BBB for many chemotherapeutics results in averting the prospective of brain cancers [7]. One approach that would circumvent the blood–brain barrier and increase drug concentration in the vicinity of the remaining GBM cells is to deliver drugs locally during or after resection surgery [8]. A concept of immediate and local delivery system would juxtapose chemotherapy drugs within the resection margin and nearby infiltrative cancer cells [9,10,11]. Based on multimodality of current treatment approaches for GBM, a single treatment is unlikely to eradicate all residual GBM cells, while the risk of recurrence remains. The risk of tumor recurrence could be diminished with the implantation of a controlled drug delivery biomaterial at the time of tumor removal surgery. A hydrogel composed of biomaterials which releases chemotherapy drugs gradually over a longer period would provide a solution. A similar strategy is currently applied in clinics, using polyanhydride and polyester-based polymer wafers delivering carmustine (BCNU) directly into the resection cavity [12,13]. The Gliadel wafer is implanted into the resection cavity following surgical removal of the tumor, and releases a drug directly into the tumor cavity, reducing the adverse side effects to healthy tissues that are commonly associated with systemic administration of chemo drugs. However, studies have shown that Gliadel only increases survival by 10–18 weeks over the placebo [14]. The moderate outcome of the Gliadel wafer might come from the rapid release of the drug and initial rapid clearance from the tumor cavity or the relatively low potency of BCNU in inhibiting proliferation and killing GBM cells.

Preclinical studies have been published which include treatment with wafers delivering temozolomide, the drug commonly used to treat GBM systemically. In an intracranial 9 L gliosarcoma model in F344 rats, it has been shown that median survival of rats treated with TMZ–BCNU loaded wafers was 50% longer than rats treated with BCNU or TMZ loaded wafers alone [15]. These promising results suggest that wafers including other chemotherapy drugs may be used as an alternative or complementary treatment to BCNU loaded Gliadel wafers and improve the clinical outcomes of GBM. However, often initially responsive GBM tumors develop resistance to BCNU and temozolomide and these treatments become ineffective due to the DNA repair mediated by the enzyme O6-methylguanine-DNA methyltransferase (MGMT) [16,17]. This acquired drug resistance leads to a high percentage of GBM recurrence and treatment failure [18].

One promising drug which is clinically used for the treatment of many malignant tumors is the anthracycline antibiotic doxorubicin (Dox) [19]. The Dox antibiotic intercalates into DNA, inhibits topoisomerase II, and induces cell death via multiple mechanisms. It is 200-fold more potent in killing GBM cell lines than BCNU and TMZ, indicating that it has a great potential in improving GBM treatment. However, when Dox is administered systemically, it induces severe side effects in healthy organs, including irreversible cardiotoxicity [20,21,22]. Furthermore, Dox is unable to cross the BBB and reach GBM tumors in concentrations sufficient to eradicate GBM tumor cells. Local drug delivery strategies have been previously explored as a means of bypassing the BBB [23]. For example, Lesniak et al. have demonstrated that local delivery of Dox using a polyanhydride polymer wafer doubled (from 21 to 45 days) the median survival of Fisher rats with intracranial glioma [24]. Similarly, in a subcutaneous flank model, it has been shown that Dox released from a polyester polymer local drug delivery system was able to slow tumor growth [25]. In both studies, the drug delivery systems displayed initial rapid drug release within the first two days. This is confirmed in other studies which include polyethylene glycol polyester polyanhydrides and other polyester drug delivery compounds. Use of polyester-based compounds may be limited, however, due to a potentially detrimental sudden burst release of the drug and possible immunogenicity or degradation products.

We have developed a biodegradable, biocompatible hydrogel formulation, based on an elastin-like polypeptide (ELP), for sustained release of Dox and local treatment of GBM. Here, ELPs are genetically engineered biopolymers, composed of repetitive units of Val-Pro-Gly-Xaa-Gly, in which Xaa is a guest residue that can be any amino acid except proline, and they possess unique features, such as elasticity, modular design, and biocompatibility [26,27,28]. The ELPs undergo inverse phase transition at an exact temperature known as inverse transition temperature (*T*_t_), below which they stay in solution and above which, due to collapse of the hydrophobic structure, they form aggregates. To selectively conjugate doxorubicin to ELP, Dox was derivatized at its C-13 keto position with a maleimidocaproyl hydrazide linker yielding (6-maleimidocaproyl)hydrazone derivative of doxorubicin (DOXO-EMCH) [29]. The acid-sensitive hydrazone linker acts as a predetermined breaking point for ensuring effective release of the drug from ELP either in the slightly acidic tumor microenvironment or after intracellular uptake in the acidic compartments of the endosomes and/or lysosomes. The ELP–DOXO aggregates are reinforced with collagen, forming biocompatible, thermo-responsive hydrogels. These ELP hydrogels may be injected into the tumor resection cavity in a liquid (sol) state using a simple syringe and needle, and then solidify (gel) in situ at body temperature.

The present study was conducted to evaluate the physical and chemical properties of ELP hydrogels, to characterize their ability to release Dox under physiologic conditions, and to investigate the effect of the fabricated hydrogels on GBM cells in vitro. By altering the final collagen concentration in the hydrogels, we adjusted ELP–collagen ratios to compare three different ELP hydrogel compositions. We measured swelling, Dox release, in vitro toxicity, and cellular drug uptake to observe and characterize any differences arising from the three formulations. These results confirm that manipulation of the ELP–collagen molar ratio alters gel microstructure and allows for tunable drug release.

## 2. Methods

### 2.1. ELP Design and Expression

The ELP-based constructs were designed and cloned via directional molecular cloning [30]. The ELP sequence selected for optimal hydrogel polymerization with collagen consists of 63 repeats of the ELP penta-peptide (Val-Pro-Gly-Xaa-Gly), where Xaa is Val, Gly, or Ala in a 5:3:2 ratio, respectively. These ELP coding sequences were modified by introducing three cysteine residues at the C-terminus of ELP for thiol–maleimide conjugation of drug molecules. All ELP constructs were expressed in the *Escherichia coli* strain BLR(DE3), using pET 25b as an expression vector. They were purified by inverse transition cycling and purity and molecular weight were confirmed by a zinc-stained single band corresponding to about 25 kDa on SDS–PAGE gel.

### 2.2. Conjugation of DOXO to ELP

Here, DOXO-EMCH, a Doxorubicin derivative (DOXO) with an acid-cleavable (6-maleimidocaproyl) hydrazone linker was covalently linked to three free cysteine residues at the carboxy terminus of ELP by thiol–maleimide coupling. To prevent spontaneous formation of disulfide bonds and to maximize efficiency of the binding process, protein conjugation with DOXO was carried out under reducing conditions. Here, ELP protein at a concentration of 100 µM was solubilized in 50 mM sodium hydrogen phosphate (Na_2_HPO_4_) buffer, pH = 7, with the addition of 10-fold molar excess (1 mM) of tris (2-carboxyethyl) phosphine (TCEP), at room temperature for 30 min. Subsequently, DOXO was added to the protein solution to a concentration of 400 µM and left to incubate for another 30 min at room temperature in the dark, followed by O/N incubation at 4 °C, protected from light. The unlabeled DOXO was removed by inverse thermal cycling. Protein concentration and labeling efficiency was calculated by measuring absorbance at 280 nm and 495 nm.

### 2.3. ELP Hydrogel Preparation

The ELP hydrogels were prepared under sterile conditions, on ice. Briefly, the acidic Collagen Type I (from rat tail, Corning, NY, USA) solution was pH-adjusted by the addition of NaOH. Salt concentration of the mixture was normalized with 10X PBS (HyClone, Logan, UT, USA), and ELP–DOXO was added to the following final concentrations: collagen (varied based on treatment) 3–6.9 mg/mL, ELP 40 μM, Dox 80 μM. The hydrogel solution was mixed slowly and consistently throughout this process, and the mixture was kept on ice to slow the polymerization process until the final concentrations of all components were reached. The hydrogel preparation process was protected from light to minimize degradation of fluorescent Dox.

### 2.4. Swelling Ratio

To measure the *swelling ratio*, ELP hydrogels were placed in PBS and, on day 8, the excess liquid was wicked away by contacting the composites (*n* = 5) with an absorbent paper for 30 s. These composites were weighed in the swollen state (*W_s_*). To obtain their dry weight (*W_d_*), the composites were placed in pre-weighed 160 μL aluminum crucibles (Mettler Toledo, Columbus, OH, USA), frozen at −20 °C, and then freeze-dried for 24 h. The swelling ratio was calculated using the following equation:$$Swelling\;ratio=\frac{W_{s}-W_{d}}{W_{d}}$$

### 2.5. Scanning Electron Microscopy

The morphology of the freeze-dried composites (*n* = 3) was studied using SEM. The composites were mounted on aluminum SEM stubs using carbon adhesive tape and sputter-coated with gold for 2 min at 9 V. The composites were observed with a SUPRA 40 scanning electron microscope (Carl Zeiss, Thornwood, NY, USA) at 5 kV using the Everhart–Thornley secondary electron detector. Images were recorded from at least two different areas on each composite at 2500× and 5000× magnifications.

### 2.6. Hydrogel Doxorubicin In Vitro Release

For in vitro studies, 10 µL of the ice-cold hydrogel solution was gently applied onto permeable cell culture inserts (Greiner Bio-one, Kremsmünster, Austria); the inserts were suspended over dry wells on a 24-well plate and hydrogels were allowed to form overnight at 37 °C in a humidified environment. The following day 500 µL PBS was added to each well, and an additional 200 µL of PBS was added to the insert chamber to ensure that the hydrogels were completely submerged and stayed in constant communication with the PBS in the wells. The PBS was withdrawn from the treated wells at indicated time points. Fluorescence from Dox was measured on a Synergy H4 plate reader (BioTek, Winooski, VT, USA), controlling for autofluorescence from PBS, and the fluorescence signal was plotted on a Dox standard to estimate drug concentration.

### 2.7. Cell Culture

The U-87 MG cells were kindly gifted from Dr. Shahn, School of Medicine, Harvard, Boston, MA. Cells were grown in Dulbecco’s modified Eagle’s minimum essential medium (DMEM) (Corning, ThermoFisher Scientific, Waltham, MA, USA), which is supplemented with 10% fetal bovine serum (FBS) (Atlanta Biologicals, Lawrenceville, GA, USA) and 1% Penicillin/Streptomycin antibiotics (HyClone, ThermoFisher Scientific). The cells were maintained at 37 °C with 5% CO_2_, and 95% humidity, in the exponential growth phase. Every two to three days cells were split using 0.05% Trypsin (HyClone, ThermoFisher Scientific).

### 2.8. Uptake and Flow Cytometry

The ELP–DOXO hydrogels with three distinct collagen concentrations were made on permeable cell culture inserts (Thincerts, Greiner Bio-one) and allowed to polymerize at 37 °C for 24 h. U-87 MG cells were plated on 24-well format at 10,000 cells per well and incubated overnight to ensure adherence. After this incubation time, permeable inserts were moved into the wells with plated cells for 4 h of treatment. Cells were then non-enzymatically stripped from the wells, washed thoroughly, and doxorubicin fluorescence was determined by flow cytometer (Gallios).

### 2.9. Localization and Confocal Microscopy

The U-87 MG cells were plated on poly-lysine treated coverslips placed in wells of a 24-well plate. Cells were allowed to adhere overnight and then were incubated with ELP–DOXO hydrogels on permeable inserts for 4 h. After treatment, coverslips were removed from the wells, washed, stained with DAPI, and affixed to slides. A Nikon confocal microscope was used to visualize fluorescence from Doxorubicin and DAPI to establish nuclear and perinuclear localization of intracellular Dox.

### 2.10. Cell Survival

For proliferation experiments, U-87 MG cells were plated on tissue culture treated 24-well plates at 6000 cells per well. Cells were allowed to adhere overnight, and the following day, ELP–DOXO hydrogels on permeable inserts were added to the wells. An additional 200 µL of DMEM was added into insert chamber to ensure hydrogel submersion and full communication with the cell culture media within plated wells. After 72 h of treatment, inserts were removed and 3-(4,5-dimethylthiazol-2-yl)-2,5-diphenyltetrazolium bromide (MTT) reagent was added [31]. The plates were incubated at 37 °C for 4 h, and then the media was aspirated. Formazan crystals were dissolved in DMSO, and absorbance was read at 570 nm (Synergy H4, BioTek) with 630 nm used as a reference. Cell survival was expressed as the percentage of absorbance signal from non-treated control wells.

## 3. Results

### 3.1. The Design and Application of ELP Hydrogels for Local Treatment of Brain Tumors

Because the surgical resection of GBM tumors is generally incomplete, tumor cells remain along the margin of the resection site. However, sol state ELP hydrogel could be injected into the resection cavity where it would solidify, forming a therapeutic hydrogel which will release a drug over a prolonged period, suppressing tumor recurrence after resection. As shown in Figure 1, ELP–DOXO and collagen are mixed, forming a composite which is initially in a liquid/sol state at room temperature. Upon injection into the tumor resection cavity, the sol state hydrogels would polymerize into a gel state at 37°. For these studies, three different formulations of ELP hydrogel were synthesized, and their properties were compared. Table 1 shows relevant composition data for each formulation which are referenced by a number that represents the molar ratio of ELP to collagen (higher numbers have more molecules of ELP per collagen molecule).

### 3.2. Scanning Electron Microscopy of Freeze Dried ELP-Based Hydrogels

Scaffold morphologies of ELP hydrogel composites formed with three different ELP–collagen molar ratios were examined by SEM. The SEM images (Figure 2) of ELP hydrogels with the lowest collagen concentration studied showed an open, porous microstructure with characteristic ELP aggregates. Hydrogels with higher collagen concentration (Gel 0.84 and Gel 0.75) showed more of dense a fibrillar and a less porous fibrillary collagenous microstructure.

### 3.3. Swelling Ratio

For ELP hydrogels to be utilized as a local drug delivery vehicle for treatment of GBM, the hydrogels must be safely injectable into the surgical resection cavity. Abrupt swelling of the hydrogel could potentially cause damage to healthy brain tissue by increasing intracranial pressure, so swelling ratio for each formulation was measured. The swelling ratio of each composite was determined by weighing hydrogels in swollen and dry states. The ELP hydrogel with the highest ELP–collagen ratio had a swelling ratio of only 8. When hydrophilic collagen content in hydrogels was increased, the swelling ratio increased roughly proportionately (Table 2).

### 3.4. In Vitro Drug Release Study

Because ELP and doxorubicin concentrations were identical in all three formulations, differences in release, uptake, and cell killing are the result of differing physical characteristics of the hydrogels based on the ELP–collagen molar ratio. To examine the release of Dox from different gel compositions, hydrogels were incubated in vitro and relative fluorescence from Dox measured at several time points, over a period of 14 days (Figure 3). Both collagen concentration and pH are expected to impact drug release from the gel, so release was monitored at physiologic pH and at lower pH, more like that of the acidic tumor microenvironment or, intracellularly, to endolysosomes. Doxorubicin release was enhanced and accelerated over this time by lower pH and lower collagen concentration. At pH 5, Gel 1.68 had released more than 70% of its total payload of doxorubicin by the end of two weeks.

### 3.5. Cellular Uptake of Dox Released from ELP Hydrogels

Release data showed that manipulation of the ELP–collagen ratio affects the rate of drug release, in a pH-dependent fashion. We next used flow cytometry to determine if the Dox released into the media would be taken up into cancer cells in vitro. The U-87 MG glioma cells were incubated with ELP–DOXO hydrogels for four hours then stripped from the cell culture plate non-enzymatically. Cellular fluorescence from Dox uptake was measured by flow cytometer, controlling for any auto-fluorescence, as shown in Figure 4. Comparatively, cells treated with hydrogels made with the lowest collagen concentration showed higher levels of uptake, and this corresponds to the release data shown in Figure 3, demonstrating faster release of Dox from compositions with more ELP per collagen.

### 3.6. Intracellular Dox Localization after Hydrogel Release

After measuring the cellular uptake of doxorubicin via flow cytometry, we next used confocal microscopy to visualize where in individual cells the drug was accumulating. Although Dox has more than one recognized mechanism of action, the best characterized requires nuclear localization of the drug molecules. The U-87 MG cells were incubated with ELP–DOXO hydrogels for four hours, stained with DAPI to label cell nuclei, and examined using a Nikon confocal microscope. Figure 5 shows that treatment with each of the three hydrogel formulations resulted in doxorubicin fluorescence co-localizing with the DAPI fluorescence of cell nuclei.

### 3.7. Cytotoxic Effect of Dox Delivered by ELP Hydrogels in GBM Cell Line

Having shown that doxorubicin was released from hydrogels in a composition-dependent manner, that the drug was taken up into the cancer cells, and that Dox was successfully localizing to the nucleus, we wanted to test the efficacy of these hydrogels in vitro. The U-87 MG cells were plated and incubated with ELP–DOXO hydrogels for 72 h, and cell survival was measured by MTT assay. Figure 6 shows the pooled results of four independent survival experiments, including measurements of doxorubicin concentration in cell culture media after 72 h of incubation. The three tested hydrogel compositions differed in efficacy, but all three successfully killed (and prevented proliferation of) more than 75% of cancer cells.

### 3.8. Cytotoxic Effect of Dox Delivered by ELP Hydrogels in U-87 MG Glioma Cell Line

The U87-GFP-fLUC cells were incubated with hydrogel for 72 h. Cell survival was determined by MTT assay and is expressed here as a percentage of the signal from non-treated control wells. These values are averages from four independent experiments. Error bars are SEM.

## 4. Discussion

Despite aggressive, multimodal treatment strategies, median survival with glioblastoma multiforme is measured in months. Complete surgical resection is rare, reoccurrence is common, and the efficacy of chemotherapy with temozolomide is limited by drug resistance and harmful off-target effects of the drug. Other chemodrugs, such as doxorubicin, are more potent against GBM cells in vitro but do not adequately traverse the BBB; additionally, Dox carries a lifetime cumulative dose limitation due to its very serious side effects. Hydrogels are emerging as promising delivery platforms for local chemotherapy because they can provide slow dissemination of drugs from the application/injection site, thus, reducing toxic systemic effects while exposing tumor tissue to the drug for a prolonged period (Sun 2020 Molecular pharmaceutics). However, although local delivery of chemotherapy maximizes drug concentration at the desired site of action, it has been shown that injections of chemotherapy drugs directly into tumor tissue had disappointing outcomes, as the drug is quickly cleared from the target area by the circulatory system [32]. In addition to burst release, hydrogels based on polyester-based composites may cause tissue toxicity due to the degradation products creating an acidic environment. To overcome these limitations, it is essential to develop biocompatible platforms for sustained drug delivery with tunable drug release and without undesirable polymer degradation by-products. One efficient and promising strategy is to use biocompatible and biodegradable hydrogels as a drug delivery system, such as ELP-based hydrogels.

These ELP hydrogels are composed of naturally occurring proteins which do not require chemical cross-linking; therefore they are biocompatible and have no toxic degradation products from the gel scaffold. Because of the phenomenon of inverse phase transition, ELPs can readily be modified on a molecular level to aggregate at a desired temperature, thus, promoting polymerization of the hydrogel under physiologic conditions. Further modification of ELP molecules allows for covalent coupling of several different chemodrug derivatives and even the possibility of loading more than one drug onto a single ELP molecule. The ELPs can also be used to deliver therapeutic peptides, so ELP hydrogels have exciting potential as a local delivery platform for varied combination treatments [33,34,35,36,37,38,39,40,41].

Here we have begun testing the suitability of ELP hydrogels for treatment of GBM, envisioning that the sol-state hydrogel could be injected directly into the cavity made by surgical resection. With this end-goal in mind, we first measured the swelling ratios of our candidate hydrogel compositions to discover what effects the implantation of ELP hydrogel might have on intracranial pressure. As predicted, increasing the concentration of hydrophilic collagen proportionally increased the measured swelling ratios. Thus, this property is also adjustable, and all values measured in these preliminary studies fall into the range of published swelling ratios of hydrogels used in in vivo experiments.

Microstructural analysis by SEM indicated that ELP and collagen are interacting with each other, and this corroborates the findings published previously showing molecular interaction between collagen and elastin and by Gurumurthy et al. showing molecular interaction between collagen and ELP [42]. Earlier attempts to create stable hydrogels with ELP alone were not successful, and we speculate that collagen functions to stabilize the scaffold and make complete polymerization more favorable. Compositions containing less than 3 mg/mL collagen did not fully polymerize; some adhesive effect from collagen appears to be necessary to produce ELP hydrogels stable enough for long-duration drug delivery.

For measurements of drug release, we covalently linked DOXO–EMCH, a doxorubicin derivative, to ELP via the drug’s acid-sensitive linker region. Here, DOXO–EMCH was designed to bind to circulating albumin and, thus, increase the half-life of Dox. The hydrazone linker region allows for cleavage of the drug from albumin (or ELP) in acidic conditions (pH below 6.5). We previously used a larger ELP construct containing a cell penetrating peptide to deliver DOXO–EMCH to tumors by utilizing externally applied, mild hyperthermia [32]. We chose DOXO–EMCH to incorporate into these early ELP hydrogels because it remains faithfully bound to ELP until it is endocytosed and subjected to the low pH of endolysosomes. This should further improve retention of the chemodrug near the site of hydrogel application. As shown in confocal localization results, Dox did accumulate in the nuclei of treated cells, supporting the theory of intracellular cleavage from ELP.

Cell uptake and proliferation experiments coupled with measurements of Dox release showed that increased concentrations of Dox released into the media corresponded with enhanced uptake and greater cell killing. Our measurement of media drug concentration was based on the fluorescence of Dox and would be unable to distinguish ELP-bound DOXO from unbound Dox, but we presume that Dox concentrations measured in the media are primarily ELP–DOXO because cleavage has been shown to be strongly pH-dependent [43]. Results of proliferation studies, showing efficacy at nanomolar concentrations, were consistent with the expectations of doxorubicin, which is similarly effective in vitro against U-87 MG [44].

Overall, these preliminary data are promising and justify further exploration of the potential of ELP hydrogels as a treatment for GBM. Further in vitro experiments are currently underway, utilizing additional GBM cell types and a larger library of ELP–collagen hydrogel formulations. Pilot animal studies are also initiated in mice with intracranially implanted GBM xenografts that have undergone partial resection of tumor followed by ELP hydrogel implantation into the resection cavity.

As we move into the reality of personalized medicine, it will be increasingly imperative to devise cancer treatments that are individualized and highly specific to the needs of each patient. Toward that end, the need for “smart”, adaptable platforms for chemotherapy delivery should be addressed urgently. The ELP hydrogels may offer a customizable, local avenue for drug administration with tunable, sustained release and little or no immunogenicity, and no toxic degradation products. These preliminary data suggest that ELP hydrogels can be engineered to follow a range of drug release profiles and could be potentially tailored to the treatment needs of cancer patients, individually.

## 5. Conclusions

Injectable ELP hydrogels laden with chemodrugs, such as doxorubicin, may present a new approach for the treatment of glioblastoma at the time of tumor resection. These hydrogels exhibit tunable, sustained drug release in physiologic conditions and could be applied locally upon tumor resection. Here, we have shown that manipulation of the ELP–collagen molar ratio impacts gel structure and swelling ratio. Additionally, a modest increase in ELP relative to collagen appears to accelerate the release of Dox, in a pH-dependent fashion. Confocal microscopy showed localization of Dox in or near the nuclei of U-87 MG cells, indicating that the acid-cleavable hydrazone linker is permitting intracellular release of Dox from ELP. The ELP hydrogels of differing ELP–collagen molar ratios effectively prevented GBM cell proliferation in vitro. These results support further preclinical studies in an animal model to verify the potential therapeutic application of ELP hydrogels for drug delivery in clinics.

## Figures and Tables

**Figure 1 pharmaceutics-14-02072-f001:**
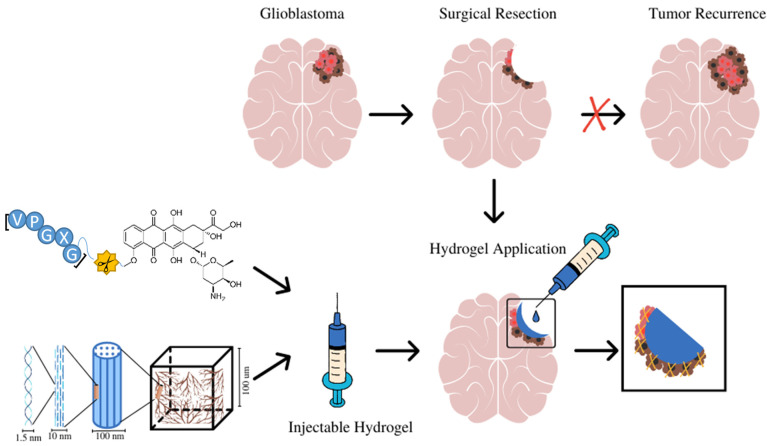
The design and application of ELP-based hydrogels for local treatment of brain tumors. ELP–DOXO hydrogels are initially in the sol state can be injected directly into the surgical resection cavity where they polymerize into a gel state and elute an anticancer drug over time, killing remaining cancer cells and preventing recurrence.

**Figure 2 pharmaceutics-14-02072-f002:**
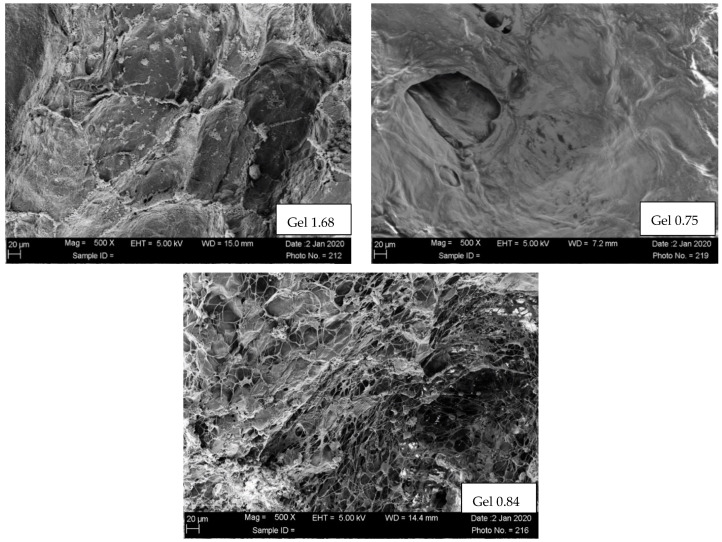
SEM of freeze-dried composites. Scaffold morphologies of ELP hydrogels examined using SUPRA 40 scanning electron microscopy (SEM, Carl Zeiss, Thornwood, NY, USA) at 5 kV. The hydrogels were freeze-dried and sputter-coated with Au/Pd for 1.5 min at 9 V.

**Figure 3 pharmaceutics-14-02072-f003:**
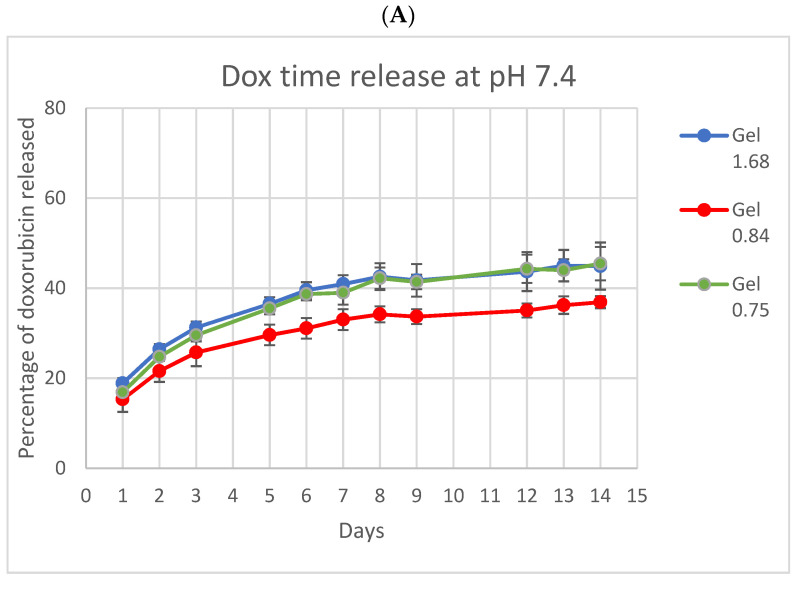
In vitro doxorubicin release from ELP hydrogels with different collagen concentrations. (**A**) At pH 7.4 the three ELP hydrogels had very similar release profiles. (**B**) At pH 5 release from the lowest collagen concentration gel was both accelerated and enhanced. Experiment was conducted at 37 °C. Error bars represent bounds of 95% confidence interval.

**Figure 4 pharmaceutics-14-02072-f004:**
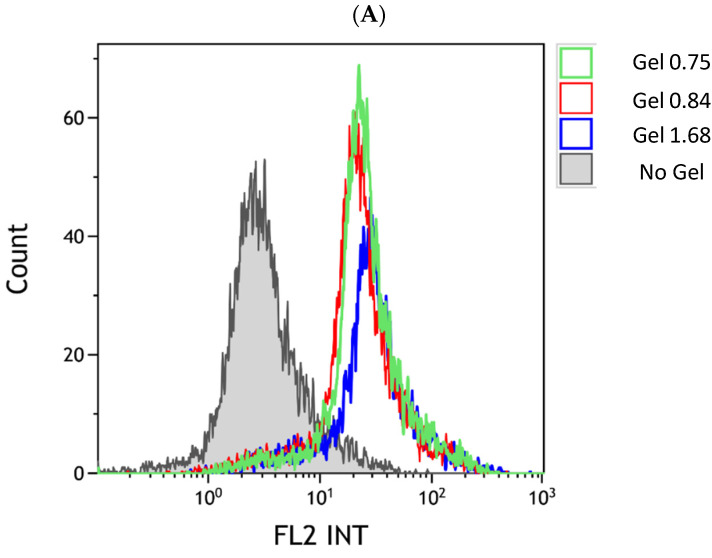
Cellular uptake of Dox released from ELP hydrogels. (**A**) The U-87 MG uptake of doxorubicin from hydrogels was measured by flow cytometry. The *x*-axis of the histogram represents fluorescence detected within the range emitted by doxorubicin; the *y*-axis represents a relative number of events/cells counted. The control peak (gray) represents cell autofluorescence. (**B**) Comparison of mean fluorescence values for three uptake experiments (including the one shown in (**A**)); bars are SEM.

**Figure 5 pharmaceutics-14-02072-f005:**
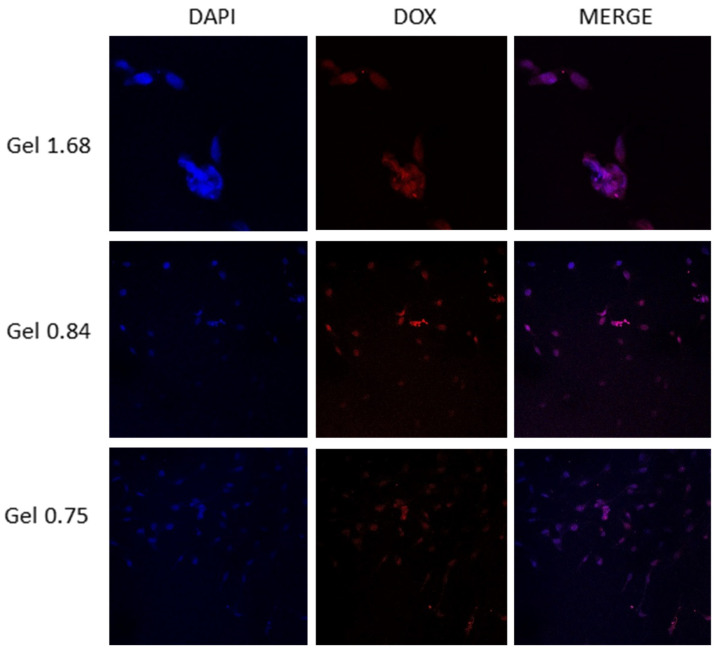
A qualitative co-localization of DAPI and doxorubicin in U-87 MG cells treated with ELP hydrogels. Cells were plated on poly-lysine treated coverslips placed in wells of a 24-well plate. These were incubated with hydrogels on permeable inserts for 4 h. After treatment coverslips were removed from the wells, washed, and stained with DAPI. A Nikon confocal microscope was used to visualize fluorescence from doxorubicin and DAPI to establish nuclear and perinuclear localization of intracellular doxorubicin.

**Figure 6 pharmaceutics-14-02072-f006:**
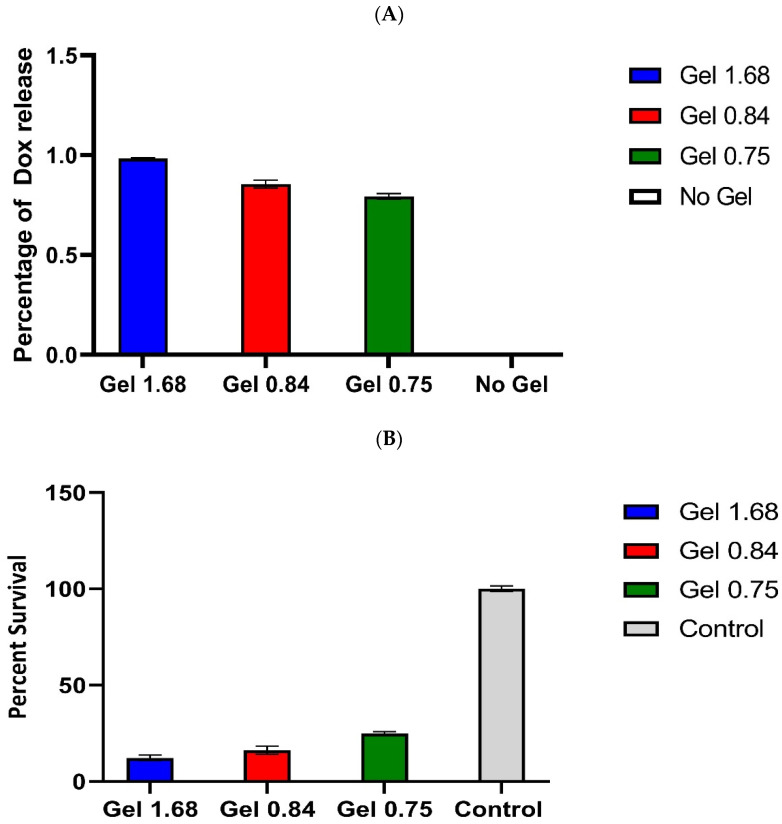
(**A**). Comparison of Dox release from different ELP based hydrogels. Doxorubicin concentration in the media of hydrogel-treated wells was approximated based on the fluorescence property of Doxorubicin. Signal values were determined using a Synergy H4 plate reader, excitation 485nm and emission 590nm, and normalized to signal from untreated media. These values were compared to a plot of a Doxorubicin standard curve and are expressed here as approximate media concentrations of Doxorubicin (µM). Error bars represent S.E.M. (**B**). Cytotoxic effect of Dox delivered by ELP hydrogels in U-87 MG glioma cell line. U87-GFP-fLUC cells were incubated with hydrogel for 72 h. Cell survival was determined by MTT assay and is expressed here as a percentage of the signal from non-treated control wells. These values are averages from four independent experiments. Error bars are S.E.M.

**Table 1 pharmaceutics-14-02072-t001:** Compositions of three ELP hydrogels compared in these studies. The designating number refers to the molar ratio of ELP to collagen (higher numbers have more molecules of ELP per collagen molecule).

ELP Hydrogel Composition		
ELP (mM)	Collagen (mg/mL)	ELP: Collagen
40	3.08	Gel 1.68
40	6.15	Gel 0.84
40	6.86	Gel 0.75

**Table 2 pharmaceutics-14-02072-t002:** Swelling ratio of hydrogels (*n* = 5) were evaluated by measuring their wet and their respective lyophilized dry weights.

	Swelling Ratio ± 95% CI
Gel 1.68	7.93 ± 5.00
Gel 0.84	21.09 ± 4.47
Gel 0.75	19.53 ± 6.22

## Data Availability

The data presented in this study are available in the article and on request from the corresponding author.

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
