# Peer review of "Elastin-like Polypeptide Hydrogels for Tunable, Sustained Local Chemotherapy in Malignant Glioma"

_pharmaceutics, 2022, doi:10.3390/pharmaceutics14102072_

Round 1
Reviewer 1 Report
Lines 45 and 59. I suggest the following revision of this line, since this is the first time the blood-brain barrier is mentioned here and it is worth introducing an abbreviation: ”…with an intact blood brain barrier(BBB),...” On line 59 I propose to remove the transcription of this term.
Line 50. The extra point “exposure. to achieve”.
Line 52. A comma instead of a point, or the word should be written with a lowercase letter. “chemotherapy, Furthermore”.
Line 62. It is necessary to add a space. “cers[6].”
Line 79. “killing GBM cancer cells.” Perhaps the word "cancer" is superfluous here, since glioblastoma can only be cancerous (tautology).
Line 89. It is necessary to add a space. “(MGMT)[15, 16].”
Line 92. It is necessary to add a space. “(Dox)[18].”
Line 99. I propose to use the already introduced abbreviation BBB. “blood brain barrier [22].”
Line 102. It is necessary to add a space. “glioma[23].”
Line 111. The common symbol for proline is Pro. “units of Val-Prol-“.
Line 138. “63 repeats of the ELP penta-peptide (VPGXG).”Here and later in the article, it is nowhere specified which amino acid is used in these "repeats". On line 112, the authors stated that "in which Xaa is a guest residue that can be any amino acid except proline,". But if the amino acids are different in the different "repeats," are the combinations of these different pentpeptides true repeats? And I would like to see a full description of the creation of the polypeptide with an indication of the pentapeptides actually used in the polypeptide.
Line 199. When writing chemical formulas, it is advisable to use lowercase. “5% CO2”.
Lines 222-223. Underlining part of the words in the chemical name. “3-(4,5-dimethylthiazol-2-yl)-2,5-diphenyltetrazolium bromide (MTT)”
Figure 1. The image of the syringe is signed "injectable hydrogel," while the text of the article rightly states that at temperatures below body temperature the solution is a sol. Perhaps we should write in the image that it is an injectable sol as well?
Line 318. The sentence here begins with a lowercase letter, not a capital letter. “cellular uptake…”
Line 363. I propose to use the already introduced abbreviation BBB.
Lines 368, 372. The references are in an improper format.
Line 386. Incorrect position of the space in the text. “treatments[31-39] .”
Note to the list of references. Authors indicate only the first or two authors and then write "et al.” I believe that the list of all authors should be given in accordance with the rules of the journal.
Author Response
We thank the reviewer for the helpful and constructive comments. We implemented all the changes and revised manuscript accordingly.

Reviewer 2 Report
The manuscript described that the potential therapeutic application of elastin-like polypeptide (ELP) hydrogels for drug delivery in clinics. The authors showed doxorubicin release from ELP hydrogels. Thus, these findings will be useful for glioblastoma treatment. Therefore, the manuscript is not too excellent to be published after revision. In other words, the manuscript is so excellent that it should be published after revision.
Comments
(1) In line of 46, can low BBB penetration be overcome by brain cancer chemotherapy through a delivery system across the blood-brain barrier into the brain based on receptor-mediated transcytosis using monoclonal antibody conjugates (Biomedicines 2022, 10, 1597. https://doi.org/10.3390/biomedicines10071597)? Why don’t refer to it, although this technology is developing?
(2) In Figure 3, in what temperature the experiments were conducted?
(3) Extracellular pH is ca. 7.4. Figure 3 showed that doxorubicin release from ELP hydrogels was at 5.0 better than at 7.4. Can doxorubicin release at 7.4 be carried out by modifying ELP hydrogels?
(4) pH in late endosomes or lysosomes is weak acid. Would doxorubicin release from ELP hydrogels be more effective in late endosomes or lysosomes than extracellular region? Can ELP hydrogels be internalized into cancer cells via endocytosis?
(5) In line of 59, “The blood brain barrier (BBB)” should be “The BBB”.
(6) In line of 46, “low BBB penetration” should be “low blood brain barrier (BBB) penetration”.
That is all.
Author Response

(The authors gave the same response as above.)
